# Neutrophil-to-Lymphocyte Ratio as an Early Predictor of Symptomatic Anastomotic Leakage in Patients after Rectal Cancer Surgery: A Propensity Score-Matched Analysis

**DOI:** 10.3390/jpm13010093

**Published:** 2022-12-30

**Authors:** Fei Tan, Kai Xu, Xinyu Qi, Pin Gao, Maoxing Liu, Zhendan Yao, Nan Zhang, Hong Yang, Chenghai Zhang, Jiadi Xing, Ming Cui, Xiangqian Su

**Affiliations:** Key Laboratory of Carcinogenesis and Translational Research (Ministry of Education), Department of Gastrointestinal Surgery IV, Peking University Cancer Hospital and Institute, Beijing 100142, China

**Keywords:** neutrophil-to-lymphocyte, anastomotic leakage, rectal cancer, propensity score-matched

## Abstract

Background: This study aimed to explore the role of postoperative neutrophil-to-lymphocyte ratio in predicting symptomatic anastomotic leakage in patients who underwent laparoscopic low anterior resection for rectal cancer. Methods: In this retrospective cohort study, we analyzed data of patients who underwent laparoscopic low anterior resection from May 2009 to May 2019. A receiver operating characteristic curve analysis was performed to evaluate the cut-off values with the best predictive efficacy of a symptomatic anastomotic leakage. In addition, a propensity score-matched analysis was performed by considering all covariate variables, and 61 patients with or without symptomatic anastomotic leakage were included in the analysis. Results: The present study included 306 patients; of these, 17 (5.56%) developed symptomatic anastomotic leakage after surgery. On postoperative day 5, compared with patients without symptomatic anastomotic leakage, those with leakage had significantly higher neutrophil-to-lymphocyte levels. Notably, a neutrophil-to-lymphocyte cut-off score of 6.54 indicated the best area under the curve of 0.818 (95% confidence interval: 0.697–0.940, *p* < 0.001) in predicting symptomatic anastomotic leakage, with a sensitivity and specificity of 76.5% and 79.4%, respectively. Conclusions: Although evidence for the predictive role of neutrophil-to-lymphocyte ratio is accumulating, it remains inconclusive. In addition, neutrophil-to-lymphocyte levels should be considered a predictive biomarker for symptomatic anastomotic leakage; however, it can more accurately be viewed as an adjunct that helps increase the clinical suspicion of emerging symptomatic anastomotic leakage.

## 1. Introduction

Colorectal cancer (CRC) is considered as one of the most common types of malignant cancer worldwide. According to GLOBOCAN 2020, it is the third most common tumor and the second most common in terms of cancer-related deaths [1]. In particular, surgical resection remains the most effective treatment for rectal cancer. Notably, anastomotic leakage (AL) is one of the most serious complications after surgery, with an incidence rate of 0–20% [2]. In addition, it has been reported that AL is associated with higher in-hospital mortality and healthcare costs [3] as well as poor oncological prognosis in long-term follow-up [4]. In particular, with the innovation of surgical instruments and the development of minimally invasive technology, the preoperative neoadjuvant therapy paves the way for low or even ultra-low anus-preservation surgery. Meanwhile, the incidence of AL has increased correspondingly. In the era of enhanced recovery after surgery or Fast Track Surgery, early rehabilitation with reduced hospital stay has been gradually accepted by surgeons [5]. However, approximately 20% of AL cases are usually diagnosed after a mean of 6–15 days of discharge [6,7]. In addition, a previous study reported that a 2.5-day delay in insufficient AL-specific treatment could increase mortality from 24% to 39% [8]. Therefore, early AL diagnosis is of great significance in clinical practice.

To date, several biomarkers of the inflammatory response, including platelet-to-lymphocyte (PLR) and neutrophil-to-lymphocyte (NLR) ratios, have been found to have the highest accuracy in predicting advanced disease among all gastrointestinal malignancies [9]. A previous high-quality meta-analysis reported that increased NLR was associated with poor overall and recurrence-free survival in patients with CRC [10]. Liu et al. reported that patients with CRC who presented with changes from a low pre-treatment NLR level to a high post-treatment NLR levels had worse overall and progression-free survival than those who presented with NLR level changes from high to low [11]. Furthermore, a previous study reported that increased preoperative NLR was associated with greater perioperative complications after colorectal surgery, with a trend toward the occurrence of AL [12]. Therefore, the predictive value of NLR in postoperative AL should be further investigated.

The present study aimed to investigate the predictive value of postoperative NLR along with PLR and lymphocyte-to-monocyte ratio (LMR) as early biomarkers of symptomatic AL in patients undergoing laparoscopic low anterior resection (LAR) for rectal cancer.

## 2. Methods

### 2.1. Patients

This retrospective study included 306 patients with pathologically confirmed primary rectal cancer who underwent laparoscopic LAR at our unit from May 2009 to May 2019. Clinicopathologic characteristics of these patients, including gender, age, body mass index (BMI), neoadjuvant therapy, American Society of Anesthesiologists (ASA) score, distance from the anal verge, smoking, alcohol consumption, history of abdominal surgery, hypertension, diabetes, pulmonary insufficiency, tumor pathology, size, differentiation and type, histopathologic staging, length of stay, preoperative hemoglobin, white blood cell count (WBC), serum albumin, and carcinoembryonic antigen (CEA), were recorded. Informed consent was obtained from each patient and their families before performing surgery. The study was conducted in accordance with the principles of the Declaration of Helsinki and approved by the Medical Ethics Committee of our hospital. This study was also reported according to the Strengthening the Reporting of Cohort Studies in Surgery criteria [13].

### 2.2. Inclusion and Exclusion Criteria

The inclusion criteria were as follows: patients aged >18 years; those with proven pathology of primary rectal cancer; those who had undergone laparoscopic LAR; those with complete clinical and pathological data; those whose blood samples were collected on postoperative days 1, 3, and 5; and those with grades B and C AL. The exclusion criteria were as follows: patients undergoing emergency surgery for perforation, bleeding, or obstruction; those with infectious or autoimmune diseases before surgery; and those with other postoperative infections, including respiratory, urinary tract, or incisional infections.

### 2.3. Surgical Procedure and Laboratory Testing

All interventions were performed by three gastrointestinal specialists at our department. In particular, the surgical procedure of laparoscopic LAR was performed, as described by Zhou et al. [14]. A pelvic drainage tube was routinely placed for all patients postoperatively at our unit. In addition, an end-to-end anastomosis was used for the colorectal anastomosis and performed using a circular stapler (ETHICONTM Circular Stapler CDH25A; Ethicon Endo-Surgery LLC, Cincinnat, Ohio, USA). The laboratory tests recorded the blood counts on postoperative days 1, 3, and 5; moreover, the NLR, LMR, and PLR were calculated.

### 2.4. Definition

This study used the standard definition of AL to ensure the accuracy of the diagnosis. Specifically, the International Study Group of Rectal Cancer recommendations of 2010 defined AL as a defect at the anastomotic site with a connection between the intra- and extra-luminal compartments [15]. Based on the severity and clinical signs, AL was further classified into asymptomatic (grade A) and symptomatic (grades B and C) types. Notably, only patients with symptomatic AL were included in the present study. AL diagnosis involves three main aspects: clinical symptoms, characteristics of the drain fluid, and enhanced computed tomography (CT) of the abdomen and pelvis. Patients with clinical symptoms, such as fever, and signs of peritonitis underwent enhanced CT for further diagnosis. Moreover, due to an unclear diagnosis, translational X-ray imaging was used for the final diagnosis. The detailed process of the procedure is as follows. First, a contrast medium (50 mL) is injected through the anus. Second, the diagnosis of AL is confirmed when the contrast agent is identified around the anastomosis or in the drainage tube under the X-ray. The follow-up time for symptomatic AL was within 1 month after surgery. In addition, the histopathological staging of rectal cancer was defined based on the Union for International Cancer Control-TNM classification (8th edition) [16]. Blood samples were obtained from the patients on postoperative days 1, 3, and 5, and in this study, detailed values of NLR, LMR, and PLR were calculated and analyzed independently.

### 2.5. Statistical Analyses

Quantitative variables were expressed as the median with a 95% confidence interval (CI), and categorical variables were expressed as total numbers and percentages. Normality of quantitative variables was assessed using the Shapiro–Wilcoxon test and further analyzed using the independent Student’s *t*-test or Mann–Whitney U-test. In contrast, categorical variables were analyzed using the chi-square or Fisher’s exact test. Notably, area under the curve (AUC), sensitivity, and specificity values based on receiver operating characteristic (ROC) curve analysis were used to analyze the variables with significant differences. Moreover, to assess the value of the multi-index combination, we used binary logistic regression to calculate the combination factor and then performed corresponding calculations using the ROC curve. The optimal cut-off score for variables was based on the Youden index, with the following grading standards: 0.51–0.60 (fail), 0.61–0.70 (poor), 0.71–0.80 (fair) 0.81–0.90 (good), and 0.91–1.00 (excellent). A propensity score-matched (PSM) analysis was performed according to the recommendations by Austin et al. to minimize the differences in baseline clinicopathological data of patients between the symptomatic AL (+) and asymptomatic AL (−) groups [17]. Notably, for the calculation of the propensity score, gender, age, and distance from the anal verge after analysis were included as confounding variables. Moreover, we used a caliper width of 0.1 and the nearest neighbor matching ratio of a 1:3 with no replacement. All variables with a *p*-value of <0.05 were considered statistically significant. Statistical analyses were conducted using the Statistical Package for the Social Sciences version 19.0 (IBM Corp., Armonk, NY, USA) and the Matching package in R, version 3.3.1 (R Foundation).

## 3. Results

### 3.1. Baseline Characteristics and Index in the Original Data

The flowchart of the study protocol is shown in Figure 1. A total of 306 patients were included in the present study; of these, 176 were men (mean age ± SD, 60.11 ± 11.26) and 130 women (mean age ± SD, 59.67 ± 11.15). Moreover, 17 (5.56%) patients who met the inclusion criteria were diagnosed with symptomatic AL; of these, 5 patients were diagnosed with grade B AL and provided conservative treatment and 12 were diagnosed with grade C AL and underwent ileostomy. The clinicopathological data of the patients are summarized in Table 1. The mean time to symptomatic AL was 5.5 (range, 4–7) days after surgery, and no deaths occurred during the perioperative period. The incidence of symptomatic AL in males was 7.5 times than that in females (88.2% vs. 11.8%), and the difference was statistically significant (*p* = 0.008). Notably, patients with symptomatic AL had a relatively lower distance from the anal verge than those without (7 (5.91–7.98) vs. 10 (8.77–9.42), *p* = 0.001). Moreover, patients who were diagnosed with symptomatic AL were younger than those who were not diagnosed with symptomatic AL (49 (44.73–58.33) vs. 61 (59.32–61.89), *p* = 0.002). Other baseline data, including BMI, neoadjuvant therapy, ASA category, smoking, alcohol consumption, history of abdominal surgery, preoperative intestinal obstruction, hypertension, diabetes, pulmonary insufficiency, preoperative hemoglobin, WBC, albumin, and CEA, as well as pathological data, including pathology type, differentiation, tumor type, and pTNM, were not significantly different between the two groups.

Changes in inflammatory indices at days 1, 3, and 5 after surgery are presented in Table 2. At postoperative day 1, no statistical significance in blood test results, except for monocytes (0.46 (0.47–0.53) vs. 0.58 (0.49–0.69), *p* = 0.025), was found between the two groups. On postoperative day 3, patients with symptomatic AL had increased median WBC (7.53 (7.44–8.19) vs. 8.66 (7.58–11.07), *p* = 0.042) and neutrophil (5.32 (5.40–6.19) vs. 6.71 (5.70–9.52), *p* = 0.042) counts than patients without. Notably, patients with symptomatic AL had a higher postoperative NLR score (8.25 (6.44–16.49) vs. 4.81 (5.58–7.13), *p* = 0.056) than patients without on postoperative day 3; however, there was no statistical significance. In addition, median WBC (6.82 (6.99–7.72) vs. 10.12 (7.85–12.23), *p* = 0.001), neutrophil (4.81 (5.02–5.71) vs. 7.25 (6.16–10.41), *p* = 0.000), lymphocyte (1.22 (1.14–1.29) vs. 0.84 (0.71–1.06), *p* = 0.004), NLR (4.16 (4.87–6.34) vs. 9.98 (7.62–14.36), *p* = 0.000), and LMR (2.13 (2.16–2.53) vs. 1.77 (1.29–2.04), *p* = 0.014) values were of striking statistical significance in patients with symptomatic AL compared to patients without symptomatic AL on day 5 postoperatively. Furthermore, ROC curve and AUC analyses for patients’ inflammatory indices on postoperative day 5 were performed as shown in Table 3 and Figure 2. The cut-off value of postoperative NLR of 6.97 presented the best efficacy (AUC: 0.802, 95% CI: 0.692–0.912, *p* = 0.000), with a sensitivity of 76.5% and specificity of 80.5%. The second predictor of symptomatic AL was postoperative neutrophils, with a cut-off value of 6.19 (AUC: 0.770, 95% CI: 0.642–0.898, *p* = 0.000) and a sensitivity and specificity of 70.6% and 73.3%, respectively. In particular, we attempted to combine the NLR and neutrophil counts for analysis in order to obtain better predictive results, but this study did not obtain better results, as expected (AUC: 0.771, 95% CI: 0.643–0.899, *p* = 0.000) (Table 3 and Figure 3). Finally, patients who developed symptomatic AL had a longer length of hospital stay than patients without (22.94 ± 7.26 vs. 15.54 ± 5.47, *p* = 0.000).

### 3.2. Matching of Covariates Using PSM Analysis

Overall, 61 patients were included in the final analysis. No statistical difference was found between the two matched groups in terms of the clinicopathological data of the patients (Table 4). Furthermore, as shown in Table 5, no statistical significance in the index scores was found at postoperative day 3, representing a large difference compared to the unpaired cohort. Notably, monocyte counts, including those of the unpaired cohort, were also statistically significant (0.44 (0.38–0.53) vs. 0.58 (0.49–0.69), *p* = 0.008) between the two groups on postoperative day 1. Consistent with previous results on postoperative day 5, the index scores, including WBC (*p* = 0.003), neutrophils (*p* = 0.001), lymphocytes (*p* = 0.015), NLR (*p* = 0.000), and LMR (*p* = 0.016), were also found to be statistically significant. Moreover, after the PSM analysis, the ROC curve analysis was performed for further evaluation. As shown in Table 6 and Figure 4, the NLR at a cut-off point of 6.54 on postoperative day 5 was demonstrated to have the optimal AUC (0.818, 95% CI 0.697–0.940, *p* = 0.000), with the sensitivity and specificity of 76.5% and 79.4%, respectively, followed by the neutrophils (AUC: 0.801, 95% CI 0.666–0.936, *p* = 0.001) at a cut-off point of 4.84. Moreover, the neutrophil and NLR values were re-analyzed after combination; however, the advantages of these combination were not seen in the analysis (AUC: 0.818, 95% CI 0.697–0.940, *p* = 0.000) as shown in Table 6 and Figure 5.

## 4. Discussion

In the present study, the incidence of symptomatic AL was 5.56%, and no in-hospital deaths caused by symptomatic AL were reported in our unit. To the best of our knowledge, this is the first study to use the PSM analysis to identify the predictive value of NLR for early prediction of symptomatic AL after laparoscopic LAR in rectal cancer. Notably, patients who developed symptomatic AL were mostly males and presented with a closer distance to the anal margin than those without AL. In addition, NLR was found to have a more accurate predictive value for symptomatic AL on postoperative day 5 with and without PSM analysis. Furthermore, NLR cut-off scores before and after PSM analysis were 6.97 and 6.54, respectively.

Previous studies have shown that a high preoperative NLR often indicates a poor prognosis [10,11], and only a few studies have reported on complications after rectal cancer surgery. First, Josse et al. reported that a preoperative NLR of ≥2.3 was associated with postoperative surgical complications and found a trend toward AL incidence without statistical significance [12]. Subsequently, Caputo et al. found that patients with a preoperative NLR above the cut-off score had significantly higher rates of postoperative complications after rectal surgery [18]. They also suggested that the cut-off values of NLR before and after neoadjuvant therapy were 2.8 and 3.8, with a reported AUC value of 0.476 and 0.692, respectively [18]. However, the present study found no statistical significance in the preoperative NLR scores between the two groups. In addition, the abovementioned two cut-off scores were relatively low, which seems unconvincing for clinical diagnosis and treatment. In addition, lower AUC values (0.476 and 0.692) reported in the two previous studies were based on lower-level statistical evidence [12,18]. Therefore, the validity of the preoperative NLR remains controversial.

There are only a few studies on the postoperative NLR level, which have not come to a consistent conclusion. In the early stages of inflammation, rapidly activated neutrophils release chemokines and cytokines while migrating to the site of inflammation; moreover, the neutrophils increase as inflammation progresses. Lymphocytes reflect the immune status of the system, and their immune activity is inhibited by bacteria and other various factors, which decrease as the inflammation progresses. However, the abovementioned phenomenon existed in a relatively late period and could not reflect the progress of the inflammation with time. Therefore, the elevated NLR levels can be used as an important marker of inflammation and are more reliable than calculating neutrophils or lymphocytes alone [19]. A retrospective study by Diana et al. with a small sample size (116 patients) reported that no statistical difference was found in the NLR levels between patients with and without AL in the first 5 postoperative days [20]. However, the largest multicenter study by Paliogiannis et al. included 1,432 patients and demonstrated that the postoperative NLR cut-off score of 7.1 showed the best efficacy in AL prediction (AUC: 0.744, 95% CI: 0.719–0.768), with the sensitivity and specificity of 72.73% and 73.44%, respectively [21]. Another study by Milk et al. reported that an NLR cut-off score of 6.5 on postoperative day 4 had a sensitivity, specificity, positive predictive value, and negative predictive value of 69%, 78%, 49%, and 88% for AL diagnosis, respectively [22]. Our study revealed that the AUC of 0.802 at postoperative day 5 below an NLR cut-off score of 6.97 was assessed as good, with a sensitivity and specificity of 76.5% and 80.5%, respectively. This value is close to the NLR cut-off score of the two previous studies by Paliogiannis et al. and Mik et al. who reported an NLR cut-off of 7.1 and 6.5 at postoperative day 4, respectively. Notably, based on the difference in test time, this closer value has a certain reference significance [21,22]. It is believed that the greater substantial growth of the postoperative NLR compared with the preoperative NLR usually indicates excessive inflammation. In particular, these inflammatory reactions can be caused by the body’s self-repair process or post-surgical complications. Therefore, postoperative NLR is believed to have some importance in the early prediction of symptomatic AL in rectal cancer.

Several other indices of inflammation, including PLR, C-reactive protein (CRP), and procalcitonin (PCT), have been reported in previous studies. Jones et al. reported that both postoperative NLR and PLR levels were associated with serious postoperative complications [23]. However, the specific cut-off scores by the ROC analysis were not reported in detail. Subsequently, Diana et al. further reported that CRP was significantly better than NLR in AL diagnosis and the best AUC was calculated using ROC curves on postoperative day 5, with a CRP value of >54 mg/dL (AUC: 0.81, sensitivity: 89%, specificity: 61%) (20). Similarly, in the study by Walker et al. involving 136 patients, they reported that CRP and NLR instead of PCT were effective predictors of anastomotic dehiscence [24]. They also found that a cut-off of CRP (105 mg/L) on postoperative day 5 demonstrated the best AUC of 0.81, with the highest sensitivity (100%) and specificity (56.5%) [24]. Notably, platelets play an important role in the body’s blood clotting function. Moreover, the tumor itself leads to an increase in the platelet count and causes the blood to become hypercoagulable, thereby leading to various postoperative complications, impairing the blood supply to tissues, and interfering with tissue healing. In the present study, the postoperative PLR value was found to have no significance in the early prediction of symptomatic AL; however, this warrants further exploration. In addition, Jabłońska B. et al. reported that lower total lymphocyte counts and serum albumin levels were found in patients who developed complications after distal pancreatectomy [25]. Moreover, this study found that patients who developed symptomatic AL had lower total lymphocyte counts and LMR levels compared to patients without AL. However, the AUC, sensitivity, and specificity values that were analyzed using the ROC curve did not yield meaningful results.

To the best of our knowledge, this is the first study to use the PSM analysis to further demonstrate the role of the NLR in early prediction of postoperative symptomatic AL after rectal surgery. However, three main limitations of the study should also be considered. The first limitation is the single-center retrospective design and the heterogeneity of the patients in this study. The second limitation is that several drugs, including nonsteroidal anti-inflammatory drugs and antibiotics, may affect blood counts and should therefore be controlled for in future studies. The third limitation is that although we used the PSM analysis, the total number of matched samples (61 patients) does not meet the ideal standard (1:3) owing to the small sample size of this study. Therefore, further prospective multicenter studies of higher quality are needed to verify the study results.

## 5. Conclusions

Although evidence for the predictive role of NLR is accumulating, it remains inconclusive. In addition, NLR should be considered a predictive biomarker for AL; however, it can more accurately be viewed as an adjunct that helps increase the clinical suspicion of emerging symptomatic AL.

## Figures and Tables

**Figure 1 jpm-13-00093-f001:**
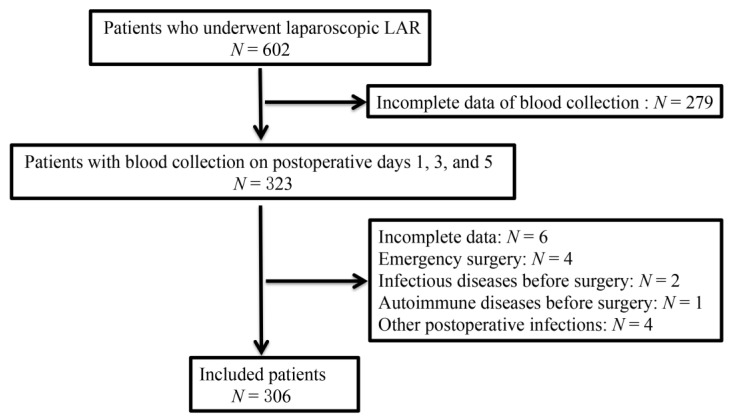
The flow diagram of study protocol.

**Figure 2 jpm-13-00093-f002:**
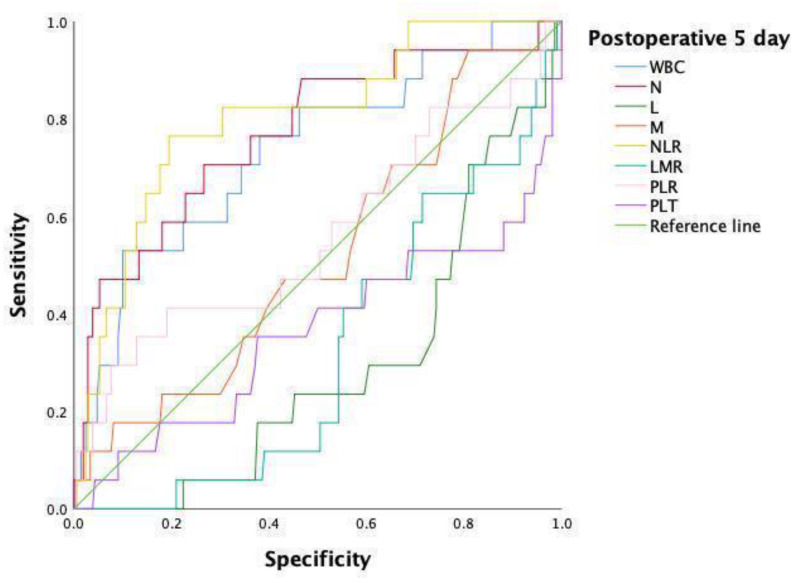
ROC curve before PSM analysis on day 5 after surgery.

**Figure 3 jpm-13-00093-f003:**
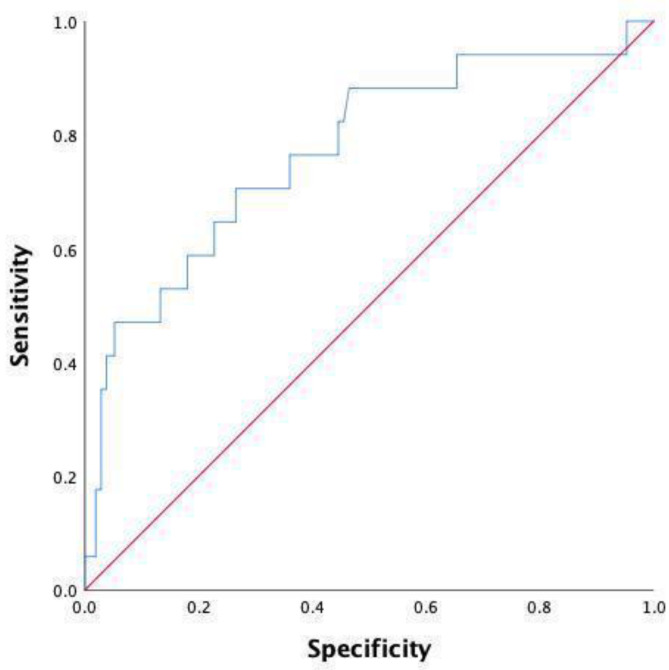
ROC curve of N and NLR before PSM analysis on day 5 after surgery.

**Figure 4 jpm-13-00093-f004:**
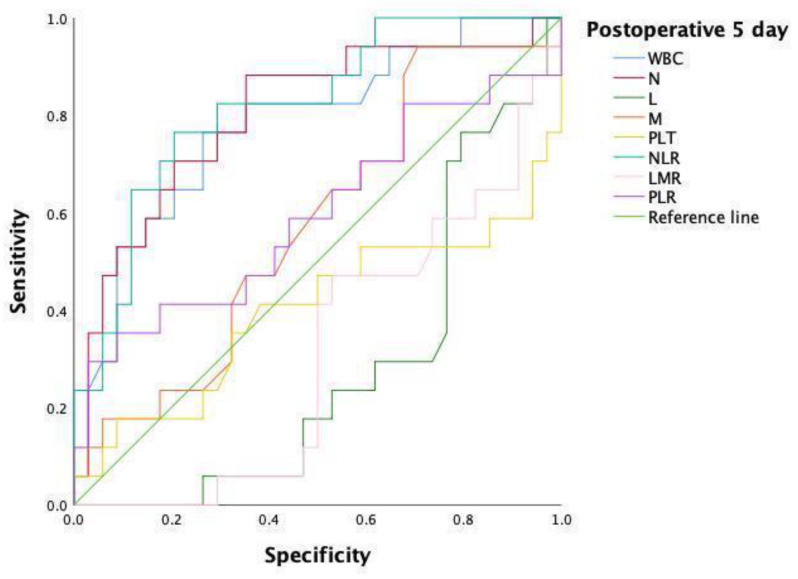
ROC curve after PSM analysis on day 5 after surgery.

**Figure 5 jpm-13-00093-f005:**
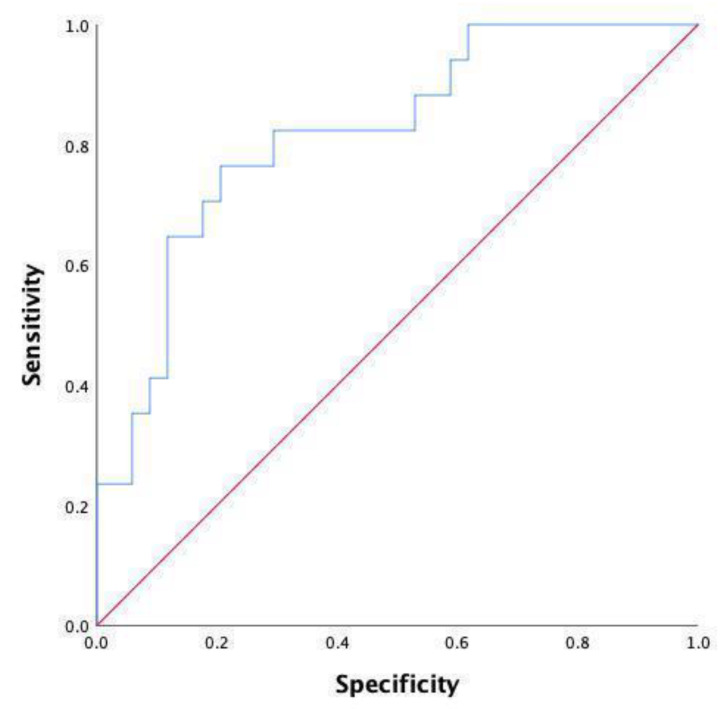
ROC curve of N and NLR after PSM analysis on day 5 after surgery.

**Table 1 jpm-13-00093-t001:** Patients clinicopathological data before propensity score-matched analysis.

Variable	All Patients	Anastomotic Leakage (N = 17)	Non-Anastomotic Leakage (N = 289)	*p* Value
Gender (n)				0.008 *
Male	176	15 (88.2)	161 (55.7)	
Female	130	2 (11.8)	128 (44.3)	
Age (median with 95% CI), years		49 (44.73–58.33)	61 (59.32–61.89)	0.002 *
Distance from the anal verge (median with 95% CI), cm		7 (5.91–7.98)	10 (8.77–9.42)	0.001 *
Body mass index (median with 95% CI) kg/m^2^		22.64 (21.85–25.69)	24.22 (23.73–24.54)	0.675
Neoadjuvant therapy (n)		4 (23.5)	47 (16.3)	0.655
American Society of Anesthesiologists score category (n)				0.746
I	43	3 (17.7)	40 (13.8)	
II	234	12 (70.6)	222 (76.8)	
III	29	2 (11.8)	27 (9.4)	
Smoking (n)		6 (35.3)	89 (30.8)	0.697
Alcohol consumption (n)		5 (29.4)	48 (16.6)	0.305
Previous history of abdominal surgery (n)		2 (11.8)	52 (18.0)	0.743
Preoperative intestinal obstruction (n)		0 (0)	29 (10.0)	0.344
Hypertension (n)		2 (11.8)	79 (27.3)	0.258
Diabetes (n)		0 (0)	34 (11.8)	0.270
Pulmonary insufficiency (n)		5 (29.4)	67 (23.2)	0.769
Preoperative hemoglobin (median with 95% CI), g/L		146 (127.61–150.74)	134 (130.3–135)	0.101
Preoperative white blood cell (median with 95% CI), 10^9^/L		6.49 (5.71–7.18)	5.98 (6.03–7.25)	0.622
Preoperative serum albumin (median with 95% CI), g/L		46.7 (40.59–47.81)	44.10 (42.63–43.91)	0.077
Preoperative CEA (median with 95% CI), ng/ml		3.13 (−3.30–25.95)	3.58 (9.60–18.65)	0.582
Duration of operation (median with 95% CI) (min)		182 (185.56–232.91)	190 (187.32–202.51)	0.204
Intraoperative blood loss (median with 95% CI) (mL)		50 (44.63–100.07)	50 (51.09–66.11)	0.075
Defunctioning stoma (n)		7 (41.2)	135 (46.7)	0.582
Pathology type of tumor (n%)				0.174
Adenocarcinoma	292	16 (94.1)	276 (95.5)	
Mucinous carcinoma	11	0 (0)	11 (3.8)	
Signet-ring cell	3	1 (5.9)	2 (0.7)	
Size of tumor (median with 95% CI), cm		3 (3.12–5.47)	4 (3.61–3.96)	0.570
Tumor differentiation (n)				0.452
Well	21	1 (5.9)	20 (6.9)	
Moderate	235	12 (70.5)	223 (77.2)	
Poor	10	1 (5.9)	9 (3.1)	
Well-moderate	7	1 (5.9)	6 (2.1)	
Moderate-poor	33	2 (11.8)	31 (10.7)	
Tumor type (n%)				0.841
Ulcer	196	10 (58.8)	186 (64.4)	
Uplift	89	6 (35.3)	83 (28.7)	
Infiltrating	21	1 (5.9)	20 (6.9)	
Pathological tumor (T) category (n)				0.294
T0	7	0 (0)	7 (2.4)	
T1	21	1 (5.9)	20 (6.9)	
T2	61	6 (35.3)	55 (19)	
T3	169	6 (35.3)	163 (56.4)	
T4	48	4 (23.5)	44 (15.3)	
Pathological node (N) category (n)				0.511
N0	172	11 (64.7)	161 (55.7)	
N1	80	5 (29.4)	75 (26.00)	
N2	54	1 (5.9)	53 (18.3)	
Pathological metastasis (M) category (n)				0.138
M0	294	15 (88.2)	279 (96.5)	
M1	12	2 (11.8)	10 (3.5)	

CEA, carcinoembryonic antigen; CI, confidence interval; * *p* < 0.05.

**Table 2 jpm-13-00093-t002:** Comparison of the mean values of the variables in patients with and without symptomatic AL before propensity score-matched analysis.

Variable	Patients	Postoperative 1 Day	Postoperative 3 Day	Postoperative 5 Day
WBC (median with 95% CI) 10^9^/L	Non-AL	8.77 (8.77–9.61)	7.53 (7.44–8.19)	6.82 (6.99–7.72)
	AL	10.06 (9.15–10.02)	8.66 (7.58–11.07)	10.12 (7.85–12.23)
	*p* value	0.054	0.042 *	0.001 *
Neutrophils (median with 95% CI) 10^9^/L	Non-AL	7.34 (7.36–8.09)	5.32 (5.40–6.19)	4.81 (5.02–5.71)
	AL	8.46 (6.80–10.53)	6.71 (5.70–9.52)	7.25 (6.16–10.41)
	*p* value	0.134	0.042 *	<0.001 *
Lymphocytes (median with 95% CI) 10^9^/L	Non-AL	0.84 (0.83–1.04)	1.08 (1.03–1.30)	1.22 (1.14–1.29)
	AL	1.01 (0.72–1.28)	0.88 (0.71–1.43)	0.84 (0.71–1.06)
	*p* value	0.505	0.314	0.004 *
Monocytes (median with 95% CI) 10^9^/L	Non-AL	0.46 (0.47–0.53)	0.51 (0.51–0.60)	0.55 (0.53–0.67)
	AL	0.58 (0.49–0.69)	0.50 (0.42–0.69)	0.53 (0.47–0.75)
	*p* value	0.025 *	0.705	0.823
Platelets (median with 95% CI) 10^9^/L	Non-AL	171 (169.67–188.06)	180 (173.21–189.71)	198 (198.98–216.22)
	AL	173 (156.04–200.20)	170 (148.77–212.28)	178 (145.19–208.70)
	*p* value	0.745	0.931	0.062
NLR (median with 95% CI) 10^9^/L	Non-AL	8.76 (9.50–11.49)	4.81 (5.58–7.13)	4.16 (4.87–6.34)
	AL	10.62 (7.76–13.67)	8.25 (6.44–16.49)	9.98 (7.62–14.36)
	*p* value	0.497	0.056	<0.001 *
LMR (median with 95% CI) 10^9^/L	Non-AL	1.78 (1.87–2.50)	2.06 (2.16–2.67)	2.13 (2.16–2.53)
	AL	1.77 (1.22–3.00)	1.60 (1.46–2.58)	1.77 (1.29–2.04)
	*p* value	0.429	0.155	0.014 *
PLR (median with 95% CI) 10^9^/L	Non-AL	200 (216.66–255.79)	159.43 (175.17–206.47)	170.68 (185.68–220.65)
	AL	187.60 (141.07–343.70)	193.18 (162.02–311.79)	163.70 (163.54–330.59)
	*p* value	0.472	0.458	0.382

WBC, white blood cells; NLR, neutrophil to lymphocyte ratio; LMR, lymphocyte to monocyte ratio; PLR, platelet to lymphocyte ratio; AL, anastomotic leakage; CI, confidence interval; * *p* < 0.05.

**Table 3 jpm-13-00093-t003:** Receiver operating characteristic curves of the variables as early predictor before propensity score-matched analysis.

Variable	AUC	95% CI	*p* Value	Cutoff	Sensitivity	Specificity
WBC	0.736	0.606–0.865	0.001 *	10.11	52.9	90
N	0.770	0.642–0.898	<0.001 *	6.19	70.6	73.3
NLR	0.802	0.692–0.912	<0.001 *	6.97	76.5	80.5
NLR and N	0.771	0.643–0.899	<0.001 *	0.07	70.6	73.5

AUC, area under the curve; CI, confidence interval; WBC, white blood cell; N, neutrophil; NLR, neutrophil to lymphocyte ratio; * *p* < 0.05.

**Table 4 jpm-13-00093-t004:** Patients clinicopathological data after propensity score-matched analysis.

Variable	All Patients	Anastomotic Leakage (N = 16)	Non-Anastomotic Leakage (N = 45)	*p* Value
Gender (n)				0.156
Male	43	14 (87.5)	29 (64.4)	
Female	18	2 (12.5)	16 (35.6)	
Age (median with 95% CI), years		49 (45.46–59.41)	51 (51.19–57.88)	0.393
Distance from the anal verge (median with 95% CI), cm		7.5 (5.99–8.14)	6 (6.21–7.48)	0.524
Body mass index (median with 95% CI) kg/m^2^		22.88 (21.91–25.95)	24.75 (23.76–25.91)	0.395
Neoadjuvant therapy (n)		3 (18.8)	14 (31.1)	0.534
American Society of Anesthesiologists score category (n)				0.500
I	7	3 (18.8)	4 (8.9)	
II	48	11 (68.7)	37 (82.2)	
III	6	2 (12.5)	4 (8.9)	
Smoking (n)		6 (37.5)	15 (33.3)	0.763
Alcohol consumption (n)		5 (31.3)	6 (13.3)	0.222
Previous history of abdominal surgery (n)		2 (12.5)	8 (17.8)	0.923
Preoperative intestinal obstruction (n)		0 (0)	1 (2.2)	1.000
Hypertension (n)		2 (12.5)	9 (20)	0.771
Diabetes (n)		0 (0)	3 (6.7)	0.560
Pulmonary insufficiency (n)		5 (31.3)	13 (28.9)	1.000
Preoperative hemoglobin (median with 95% CI), g/L		143 (126.36–151.01)	132 (123.10–134.90)	0.064
Preoperative white blood cell (median with 95% CI), 10^9^/L		6.58 (5.74–7.28)	5.73 (3.92–11.27)	0.121
Preoperative serum albumin (median with 95% CI), g/L		46.60 (40.11–47.72)	45.40 (40.72–45.15)	0.446
Preoperative CEA (median with 95% CI), ng/ml		2.94 (−4.86–26.34)	3.42 (0.35–25.55)	0.909
Duration of operation (median with 95% CI) (min)		200 (195.90–243.60)	200 (190.90–240.35)	0.251
Intraoperative blood loss (median with 95% CI) (mL)		50 (44.25–103.25)	50 (47.39–97.94)	0.389
Defunctioning stoma (n)		6 (32.5)	29 (64.4)	0.061
Pathology type of tumor (n%)				0.459
Adenocarcinoma	59	15 (93.8)	44 (97.8)	
Mucinous carcinoma	1	0 (0)	1 (2.2)	
Signet-ring cell	1	1 (6.2)	0 (0)	
Size of tumor (median with 95% CI), cm		3.50 (3.22–5.66)	4.0 (3.22–3.94)	0.275
Tumor differentiation (n)				0.632
Well	3	1 (6.2)	2 (4.4)	
Moderate	45	11 (68.7)	34 (75.6)	
Poor	2	1 (6.2)	1 (2.2)	
Well-moderate	2	1 (6.2)	1 (2.2)	
Moderate-poor	9	2 (12.7)	7 (15.6)	
Tumor type (n%)				0.539
Ulcer	39	9 (56.3)	30 (66.7)	
Uplift	16	6 (37.5)	10 (22.2)	
Infiltrating	6	1 (6.2)	5 (11.1)	
Pathological tumor (T) category (n)				0.480
T0	2	0 (0)	2 (4.4)	
T1	2	1 (6.2)	1 (2.2)	
T2	20	5 (31.3)	15 (33.3)	
T3	28	6 (37.5)	22 (49)	
T4	9	4 (25)	5 (11.1)	
Pathological node (N) category (n)				0.413
N0	34	10 (62.5)	24 (53.4)	
N1	16	5 (31.3)	11 (24.4)	
N2	11	1 (6.2)	10 (22.2)	
Pathological metastasis (M) category (n)				0.841
M0	56	14 (87.3)	42 (93.3)	
M1	5	2 (12.7)	3 (6.7)	

CEA, carcinoembryonic antigen; CI, confidence interval.

**Table 5 jpm-13-00093-t005:** Comparison of the mean values of the variables in patients with and without symptomatic AL after propensity score-matched analysis.

Variable	Patients	Postoperative 1 Day	Postoperative 3 Day	Postoperative 5 Day
WBC (median with 95% CI) 10^9^/L	Non-AL	8.71 (8.02–9.65)	6.81 (6.54–8.59)	6.27 (6.00–8.00)
	AL	10.06 (9.15–10.02)	8.66 (7.58–11.07)	10.12 (7.85–12.23)
	*p* value	0.187	0.146	0.003 *
Neutrophils (median with 95% CI) 10^9^/L	Non-AL	7.38 (6.78–8.33)	5.18 (4.74–6.80)	4.52 (4.16–5.95)
	AL	8.46 (6.80–10.53)	6.71 (5.70–9.52)	7.25 (6.16–10.41)
	*p* value	0.202	0.061	0.001 *
Lymphocytes (median with 95% CI) 10^9^/L	Non-AL	0.79 (0.71–0.92)	1.09 (0.92–1.25)	1.28 (1.02–1.42)
	AL	1.01 (0.72–1.28)	0.88 (0.71–1.43)	0.84 (0.71–1.06)
	*p* value	0.081	0.545	0.015 *
Monocytes (median with 95% CI) 10^9^/L	Non-AL	0.44 (0.38–0.53)	0.48 (0.43–0.63)	0.52 (0.44–0.64)
	AL	0.58 (0.49–0.69)	0.50 (0.42–0.69)	0.53 (0.47–0.75)
	*p* value	0.008 *	0.816	0.372
Platelets (median with 95% CI) 10^9^/L	Non-AL	167 (145.63–175.49)	171 (151.51–183.77)	189 (176.05–211.79)
	AL	173 (156.04–200.20)	170 (148.77–212.28)	178 (145.19–208.70)
	*p* value	0.559	0.828	0.394
NLR (median with 95% CI) 10^9^/L	Non-AL	8.82 (8.41–11.64)	5.17 (4.62–7.93)	4.24 (3.58–6.44)
	AL	10.62 (7.76–13.67)	8.25 (6.44–16.49)	9.98 (7.62–14.36)
	*p* value	0.679	0.219	<0.001 *
LMR (median with 95% CI) 10^9^/L	Non-AL	1.79 (1.45–2.83)	2.06 (1.81–2.78)	2.31 (2.05–3.22)
	AL	1.77 (1.22–3.00)	1.60 (1.46–2.58)	1.77 (1.29–2.04)
	*p* value	0.717	0.532	0.016 *
PLR (median with 95% CI) 10^9^/L	Non-AL	200 (180.92–257.88)	150.44 (144.47–202.58)	152.87 (151.64–213.23)
	AL	187.60 (141.07–343.70)	193.18 (162.02–311.79)	163.70 (163.54–330.59)
	*p* value	0.936	0.663	0.244

WBC, white blood cells; NLR, neutrophil to lymphocyte ratio; LMR, lymphocyte to monocyte ratio; PLR, platelet to lymphocyte ratio; AL, anastomotic leakage; CI, confidence interval; * *p* < 0.05.

**Table 6 jpm-13-00093-t006:** Receiver operating characteristic curves of the variables as early predictor after propensity score-matched analysis.

Variable	AUC	95% CI	*p* Value	Cutoff	Sensitivity	Specificity
WBC	0.779	0.641–0.916	0.001 *	7.40	76.5	73.5
N	0.801	0.666–0.936	0.001 *	4.84	88.2	64.7
NLR	0.818	0.697–0.940	<0.001 *	6.54	76.5	79.4
NLR and N	0.818	0.697–0.940	<0.001 *	0.29	76.5	79.4

AUC, area under the curve; CI, confidence interval; WBC, white blood cell; N, neutrophil; NLR, neutrophil to lymphocyte ratio; * *p* < 0.05.

## Data Availability

The data presented in this study are available upon request from the corresponding author according to “MDPI Research Data Policies”.

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
