# Peer review of "Neutrophil-to-Lymphocyte Ratio as an Early Predictor of Symptomatic Anastomotic Leakage in Patients after Rectal Cancer Surgery: A Propensity Score-Matched Analysis"

_jpm, 2022, doi:10.3390/jpm13010093_

Round 1

Reviewer 1 Report

In this study, Fei Tan, et al. explored the role of postoperative neutrophil-to-lymphocyte ratio in predicting symptomatic anastomotic leakage in patients who underwent laparoscopic low anterior resection for rectal cancer. The authors conclude that the neutrophil-to-lymphocyte on postoperative day 5 is an important predictor of symptomatic anastomotic leakage, and blood counts are worthy to be considered in clinical practice. The topic of this paper is current and clinically relevant in the management of anastomotic leakage in patients who underwent laparoscopic AL for rectal cancer. 

Few suggestions to consider

-       Consider including a flow diagram of study protocol

-       Authors report very high incidence of symptomatic AL in males. How would they explain this finding?

-       Consider adding mean NLR for various tumor stages.

-       How NLR was calculated?

-   How many physicians performed the surgeries during the study period?

-       Consider adding following operative data if available: OR time, Blood loss, blood transfusion, Albumin level, and Hb level.

Author Response

Dear reviewers,

    Thank you for your dedication for our study and please see the attachment. 

    Merry Christmas for you!

Sincerely yours

Doctor_Xinyu Qi

Reviewer 2 Report

It is an interesting manuscript covering an emerging clinical risk factor NLR.

A few comments should be addressed prior to reconsidering this manuscript for publication.

1 - English language requires editing, a few mistakes both spelling and grammar have been spotted throughout the text

2 - Please present the data as median and 95% CI rather than mean with SD/SE

3 - Please increase the resolution of ROC graphs

4 - Since you have collected so many clinical parameters please consider regression analysis to elucidate the prognostic relevance of other markers too and perhaps to double check NLR ratio

5 - Discussion chapter is fine, but if you add a few more sentences on potential mechanisms of such association it will be very much appreciated by reader.

Author Response

(The authors gave the same response as above.)

Reviewer 3 Report

I would like to thank you for the opportunity to review the original study entitled "Neutrophil-to-lymphocyte ratio as early predictor of symptomatic anastomotic leakage in patients after rectal cancer surgery: 3 A Propensity score-matched analysis" by Tan et al. The authors of the study performed a well thought-out analysis on a the role of NLR as a predictor of poor outcomes after colorectal surgery. This topic has gained much attention in recent years, yet NLR still hasn't found its place in routine clinical practice, as evidence remains inconclusive.

In the current study, the authors performed a retrospective PSM study that complements nicely the existing literature and fits well within the scope of the journal. The methods utilized are appropriate and the authors did well to acknowledge the limitations of the study and contrast their results with those previously reported elsewhere. I only have a few minor remarks:

1.  Overall the language of the text is good, I would suggest a careful rereading to correct the few mistakes that are present.

2. In the abstract, results subsection: I suggest that the authors present the findings of the matched cohort instead of the unmatched one (reportedly, a cut-off value of 6.54, AUC 0.81).

3. page 3, line 56 "primary rectal cancer who adopted laparoscopic LAR at our unit": consider changing adopted to underwent.

4. In the results section of the manuscript please consider providing data on the postoperative day on which the anastomotic leaks were diagnosed (as median and range). While it is certain that most leaks occur late in the postoperative course, early leaks (such as those occurring on or before the fifth postoperative day) cannot be incorporated in the present analysis.

5. page 4, line 87 "transanal X-ray imaging": I wonder what the authors mean by this. Please consider a more accurate phrasing.

6. page 8, lines 130-131 "Without statistical significance, patients with symptomatic AL had a significantly increased postoperative NLR value: please rephrase to "had an increased postoperative NLR" (remove significantly).

7. page 11, lines 159-161: These first two sentences of the PSM analysis are redundant and should be removed.

8. page 11, line 163 "which further proves that the PSM is running well": this also redundant, consider removing.

9. The discussion is well structured and concise. I believe a short paragraph on the rationale of why postoperative day 5 is the optimal timing of measuring the NLR (as opposed to postop day 3 or 4) would strengthen the paper.

10. Conclusions "Therefore, NLR is an important predictor of symptomatic AL after laparoscopic LAR for rectal cancer": while evidence on the predictive role on NLR are accumulating, it is still not conclusive. Although NLR merits consideration as a predictive biomarker for AL, it may be more accurately thought of as an adjunct that serves to heighten clinical suspicion for an emerging anastomotic leak. Therefore, I suggest changing the conclusion of the manuscript to better reflect the actual role of NLR in clinical practice.

11. Title: change to "as an early predictor".

Author Response

Dear reviewers,

    Thank you for your dedication for our study and please see the attachment. 

    Merry Christmas for you!

Sincerely yours

Round 2

Reviewer 2 Report

Authors have amended article according to my previous comments. I still believe the the presentation of data in median and 95% CI will be more trustworthy.

Nonetheless, the paper looks good now and I am happy to endorse it for publication. However, I recommend the editorial board to do an intensive check of English language, there are still some spelling mistakes spotted, although much better than the previous time. thank you.